# Material Characterization of PMC/TBC Composite Under High Strain Rates and Elevated Temperatures

**DOI:** 10.3390/ma13010167

**Published:** 2020-01-01

**Authors:** Przemysław Golewski, Alexis Rusinek, Tomasz Sadowski

**Affiliations:** 1Faculty of Civil Engineering and Architecture, Lublin University of Technology, Nadbystrzycka 38, 20-618 Lublin, Poland; pgolewski@gmail.com; 2Laboratory of Microstructure Studies and Mechanics of Materials, UMR-CNRS 7239, Lorraine University, 7 rue Félix Savart, BP 15082, CEDEX 03, 57073 Metz, France; alexis.rusinek@univ-lorraine.fr

**Keywords:** polymer-matrix composites (PMCs), thermal barrier coating (TBC), split Hopkinson pressure bar, carbon fiber reinforced polymer (CFRP)

## Abstract

Polymer matrix composites (PMC), despite their many advantages, have limited use at elevated temperatures. To expand the scope of their uses, it becomes necessary to use thermal barrier coatings (TBC). In addition to elevated temperatures, composite structures, and thus TBC barriers, can be exposed to damage from impacts of foreign objects. Therefore, before using the thermal barrier in practice, knowledge about its behavior under high-speed loads is necessary. The paper presents results for samples with the PMC/TBC system subjected to dynamic compression using a split Hopkinson pressure bar (SHPB). The substrate was made of CFRP (carbon reinforced polymer) with epoxy matrix and twill fabric. TBC was made of ceramic mat saturated by commercial hardener from Vitcas company. The tests were carried out at ambient temperature and elevated temperature—55 °C and 90 °C. Tests at ambient temperature were carried out for three pressure levels: 1, 1.5, and 2 bar. Only the pressure of 1 bar was used for the elevated temperature. Studies have shown that the limit load is 1 bar for ambient temperature. At 1.5 bar, cracks occurred in the TBC structure. Increased temperature also adversely affects the TBC barrier strength and it is damaged at a pressure of 1 bar.

## 1. Introduction

Thermal barrier coatings [1] have been used for many years in the aviation industry, mainly in turbine engine designs. They are used for coating working and stationary turbine blades, combustion chambers, and exhaust system components. To reduce the weight of the aircraft, the goal is to replace metal parts with parts made of polymer matrix composites, which have much better weight to strength ratios. However, their main disadvantage is the relatively low operating temperature, which depending on the matrix, can be in the range of 120 (epoxy matrix) [2] to 250 °C (BMI matrix) [3]. In contrast, temperatures occurring in hot engine parts are often in the range of 800–1000 °C, so it is necessary to use protective barriers and internal cooling. A popular protective barrier is yttria-stabilized zirconia (YSZ) applied by the air plasma spray (APS) method on the MCrAlY bond coat, which is produced by the high velocity oxy fuel (HVOF) method on a metal substrate. Both the APS and HVOF methods cannot be directly applied to a substrate made of polymer matrix composites. In the first case, the substrate will be damaged as a result of high plasma temperature, while in the second one, the surface will erode instead of depositing the coating. There are other methods, such as physical vapor deposition (PVD) or sol-gel, but they are dedicated to relatively small parts. The cold spray technique is a promising method [4], but using it, only metallization can be made by producing an interlayer. In [5], the authors proposed a different solution based on the use of a ceramic mat that was applied to the CFRP (carbon reinforced polymer) substrate in one process during curing of the composite. The tests confirmed its effectiveness in terms of thermal insulation; however, the ceramic mat had almost zero mechanical strength. Therefore, in subsequent works, the authors proposed its strengthening by saturation with a hardener using water glass in its composition. Thermal and static tests were also carried out [6]. In this work the dynamic and thermo–dynamic loads were considered for this type of material.

Dynamic loads to which thermal barriers in aviation are exposed may appear as a result of sucking foreign objects into the engine; e.g., grains of sand, dust, or debris. In the worst case, hitting the protective coating may cause its erosion or surface damage, causing cracks [7,8]. These are very complex phenomena; however, in order to build a numerical model, material data is necessary regarding the behavior of the protective coating for different strain rates and temperatures. For this purpose, the SHPB device was used in the work to reach high strain rates. Tests were carried out for three strain rate levels (≈1500, ≈2000, ≈2500 l/s) and three temperatures (room, 55 °C and 90 °C).

## 2. Research Using SHPB

The SHPB device can be used in practice for any type of material, from very hard and durable ones, such as Al_2_O_3_ sintered ceramics, to metals, and even soft materials, such as polymers and even foams. However, the right choice of rod material is required for each of them. Dynamic joint shearing or bending tests are also possible [9]. In case of the PMC samples testing, steel bars are used, while the samples have a circular or square cross-section. There is currently no standard for sample dimensions; the following dimensions are used for square samples: 13 × 13 × 10 mm [10], 8.5 × 8.5 × 2 mm [11], 7 × 7 × 5 mm [12], 10 × 10 × 10 mm [13], and many others. Polymer matrix composites have orthotropic properties; therefore, often two or three main directions are tested, which is also done when the main direction does not coincide with the load direction [14]. The purpose of the tests, in addition to determining the elastic characteristics, is also to assess the limit value of strain rates at which the material is not damaged. The tests are carried out for different strain rates; e.g., for three values as in [15] to nine values as in [10]. For each orientation, different damage models can be distinguished; e.g., (1) crushing of the resin, (2) formation of damage zone (V shape) (3) macro cracking, (4) micro cracks, and (5) propagation and failure [16]. Often, delamination and buckling of the fibers and the sample fragmentation appears [17]. The buckling phenomenon can be eliminated by additional reinforcement towards the z axis—an axis perpendicular to the layer system. However, this requires the use of special fabrics (3D braided) and the use of the VARTM (vacuum assisted resin transfer molding) method [18,19,20,21,22,23,24,25,26,27]. In addition, the effects of the strain rate, aging, and temperature are also considered. In [3], oxidation samples were subjected to elevated temperatures of 195 and 245 °C for two months. A BMI matrix was used, whose working temperature is 204 °C. The tests were conducted for one velocity of 200 1/s. For the samples aged at 245 °C, the elastic modulus decreased by 45.3%, 58.6%, and 54.5%, whereas for ageing temperature equal to 195 °C, the decreases were 1.92%, 6.83%, and 39.7% for the three directions. Composites and their thermal barriers may also be exposed to low temperatures in such structures as liquid hydrogen tanks in rockets or elements of satellite structures in open space. In [26], tests were conducted for cryogenic temperatures: 26, −50, −100, −140 °C and velocities in the range 1300–2100 1/s. For low temperatures, the modulus of elasticity and maximum stress increase, but the strains on damage level are the same. Research using SHPB for fiber composites can also be supported by a thermal imaging camera [28,29] and a DIC (digital image correlation) system [21]. However, the most valuable information is provided by true stress—true strain graphs and microscope images observations. 

In the works analyzed, no results were found regarding thermal barrier coating (TBC) protective barriers used on composites or double-layer materials with different mechanical properties. Therefore, our following studies will fill this gap.

## 3. Materials and Methodology

The test samples (Figure 1) made of two materials, PMC substrate and TBC layer, were prepared in two stages.

The mechanical strength of the ceramic mat is almost zero, hence the first stage was its hardening. Both the 3 mm thick ceramic mat and hardener came from one supplier—Vitcas company (Bristol, United Kingdom). The hardening of the mat was done manually using a rubber roller. The hardener was added gradually and uniformly distributed over the entire surface. After complete saturation, the 200 × 300 mm mat was allowed to dry freely, turning it over several times. The dried mat is characterized by much greater stiffness and hardness compared to the mat before hardening. Its parameters were determined in [6].

Nowadays, in the aviation, automotive or yacht industry, prepregs are used to make composites. Their curing usually takes place in an autoclave. Hence, in the second stage, a package was created consisting of four layers of Gurit EP121 prepreg and a layer of previously hardened ceramic mat. Both materials were sealed in a vacuum bag and cured in an autoclave. The advantage of this process was to obtain smooth and parallel surfaces of the composite substrate and protective layer. This is important because tests using SHPB require that the surfaces adhere evenly to both bars. The last stage of sample preparation was their cutting using a CNC plotter.

Each sample was accurately measured so as to be able to enter geometry into the WASP (Waves Analysis and Study Program). The tests were carried out on a total of 18 samples; their detailed characteristics are shown in Table 1.

Static tests were carried out on an MTS testing machine with a 100 kN measuring head. The SHPB was equipped with steel bars with a diameter of 20.5 mm. Both the input bar and output bar were 1.5 m long. The length of the projectile, allowing to reach a pulse equal to (1) was 0.4 m.
(1)$$t=\frac{2l_{p}}{C_{b}}$$
where: *t*—pulse duration [s], $l_{p}$—length of projectile [m], and $C_{b}$—velocity of longitudinal sound wave propagation in the bar [m/s].

For testing at elevated temperatures, a special furnace was used, enabling both sample heating and control by measurement using a thermocouple (Figure 2). After reaching the set temperature *T*_0_, the furnace controller continues to maintain it for a programmed period of time to achieve a homogeneous temperature in the sample. To avoid a gradient of *T*_0_, end of the bars are heated also.

After dynamic impact tests, the following results were obtained (Figure 3a). For further processing of it, the charts require proper cutting, as shown in Figure 3b. This was done in the DIAdem program.

Further processing of the results was performed using the WASP program. Based on the results of incident wave $\varepsilon_{I}(t)$, reflected wave $\varepsilon_{R}(t)$*ε*_R_(*t*), and transmitted wave *ε*_T_(*t*) (Figure 3), it is possible to determine a dynamic curve of material strengthening of a sample. The speed of the contact surface $V_{1}\left(t\right)$ of the input bar is expressed by the equation:(2)$$V_{1}\left(t\right)=C_{b}\left(\varepsilon_{I}-\varepsilon_{R}\right)$$
while the velocity of the front of the output bar $V_{2}\left(t\right)$ describes the formula:(3)$$V_{2}\left(t\right)=C_{b}\varepsilon_{T}$$
therefore, the average strain rate of the sample is determined by the equation:(4)$$\dot{\varepsilon }\left(t\right)=\frac{V_{1}-V_{2}}{l_{s}}=\frac{C_{b}}{l_{s}}\left(\varepsilon_{I}-\varepsilon_{R}-\varepsilon_{T}\right)$$
where $l_{s}$ is the initial length of the sample.

Nominal deformation of the sample during the experiment ε(t) is obtained by integrating Equation (4) and has the form:(5)$$\varepsilon \left(t\right)=\frac{C_{b}}{l_{s}}\int_{0}^{t}\left(\varepsilon_{I}-\varepsilon_{R}-\varepsilon_{T}\right)dt$$

The values of normal forces occurring on the contact surfaces of bars can be calculated from the following equations:(6)$$F_{1}\left(t\right)=E_{b}A_{b}\left(\varepsilon_{I}+\varepsilon_{R}\right)$$
(7)$$F_{2}\left(t\right)=E_{b}A_{b}\varepsilon_{T},$$
where $E_{b}$ is the Young modulus of the bars and $A_{b}$ is the cross-sectional area of the bars.

Nominal stresses σ(t) in the sample during deformation is given by formula:(8)$$\sigma \left(t\right)=\frac{F_{1}\left(t\right)+F_{2}\left(t\right)}{2A_{s}}$$

After substitution of Equations (6) and (7) into Equation (8), we obtain:(9)$$\sigma \left(t\right)=\frac{E_{b}}{2}\frac{A_{b}}{A_{s}}\left(\varepsilon_{I}+\varepsilon_{R}+\varepsilon_{T}\right)$$
where $A_{s}$ is the initial cross-sectional area of sample. 

## 4. Results and Discussion

### 4.1. Static Tests

The samples were characterized by a large difference in the modulus of elasticity of the substrate and the protective layer, while the PMC material was more than 10 times stiffer. Thus, the deformations that arise in the sample mainly refer to the material of the hardened ceramic mat. The tensile strength of 12 MPa was achieved in the uniaxial tensile test [6], and while no plastic deformation of the sample was observed, there was the so-called brittle cracking.

As a result of static compression at a constant strain rate of 4.16 × 10^−3^ 1/s, the engineering stress–strain graph as in Figure 4 was obtained for three samples 1_1, 1_2, and 1_3. It should be noted that for PMC samples, graphs were obtained in the form of a straight line, e.g., [15], whereas the shape of the graph obtained in current tests can be divided into several stages.

In the first stage, we can observe a linear response. The limit value is on the level of 100 MPa. So, just like for brittle materials, there is a large difference between tensile and compressive strength.

The second stage is characterized by the curvature of the graph. The microscopic tests carried out in [6] proved that it is due to compaction of pores in the composite structure. Therefore, the process of closing them follows.

After complete closing of the pores, there is a linear increase in stress—stage III. In stage IV, the TBC layer is squeezed out of the sample area, stresses increase even faster, as the volume of the rigid composite substrate begins to dominate. Static compression was carried out to a displacement of 2 mm. Figure 5 shows the view of the sample after the compression test.

One can see no cracks in the middle of the sample. There is also no damage in the PMC substrate layer. However, symmetrical areas of extruded TBC material are visible, which have been additionally subjected to numerous cracks.

### 4.2. Dynamic Tests at Ambient Temperature

In the next stage, dynamic tests were carried out using the SHPB device for three loads at ambient temperature. The graphs shown in Figure 6 were analyzed using WASP. The following conclusions can be drawn from the analysis of the graphs:With increasing impact velocity, repeatability of results is higher; however, to prove this thesis, it is necessary to perform tests on series with greater numbers of samples. Of course, there is no claim that the differences in the graphs are in the last stage when the sample is destroyed in an uncontrolled manner. However, for sample 1 and load 1 bar, and sample 2 and load 2 bar, there are differences in the first stage of loading.As the velocity of impact increases, the elastic limit increases: The elastic limit can be taken as the end of a straight section. It is especially noticeable for 1 bar and 2 bar loads. The elastic limits determined on the basis of the graphs are: 100 MPa, 160 MPa, and 220 MPa respectively, for 1, 1.5, and 2 bar loads or corresponding strain rates ≈1500, ≈2000, and ≈2500 1/s. The dynamic modulus, whose determination methods are presented in [30], was also increasing.There is a limit value for dynamic strength: As the load velocity increases, the maximum stress value increases from 350 MPa for a load of 1 bar to 400 MPa for a load of 1.5 bar. A similar value was also obtained for a 2 bar load.The nature of the damage can be determined from the shape of the graph.

In the tests, the loads were selected in such a way as to obtain three different effects, visible both on the graphs and on the sample microscopic observations. Taking into account part of the graph after reaching the maximum stress, for a load of 1 bar we obtained a relatively long flat section; for a load of 1.5 bar, a slight curvature (stress drop); and for a load of 2 bar, the stress decreased linearly and rapidly.

The effects described are reflected in the microscopic observations of samples after testing. For a 1 bar load (Figure 7), none of the samples were damaged. There are also no visible cracks or plastic deformations. Thus, the loads applied were safe.

Figure 8 presents photos for all samples in the series for a load of 1.5 bar. In this case, sample 3_2 was damaged on the perimeter, but the TBC barrier remained consistent with the substrate. The remaining samples 3_1 and 3_3 showed numerous cracks, especially at the edges. The results show that a load limit value that must not be exceeded was reached.

Figure 9 provides information that further increasing of the load will result in almost complete removal of the TBC layer from the substrate. Only small fragments, concentrated around the sample axis, still remain.

The results can also be divided into two groups, as the authors do in the paper [17]. We can distinguish a non-damaging test (1 bar) and damaging tests (1.5 and 2bar).

### 4.3. Dynamic Tests at Elevated Temperatures

The tests carried out in [6] on the PMC/TBC system at thermal loads at 850 °C showed effective thermal barrier operation. However, there is no information regarding the strength of this type of coating at thermal and mechanical loads caused by high strain rate. Literature analysis shows that tests at non-room temperatures and SHPB are rarely carried out for PMC materials.

In the work [31] the influence of reinforcement and high temperature was examined. Two types of additives were used in the form of spheres (spherical particles) and milled fibers. Samples were heated to 180 °C, and then cooled and tested at room temperature. Milled fibers affect the increase of stress; however, the energy absorption is lower than in the case of spherical additives. Heating above the glass transition temperature (*T*_g_) improves dynamic properties in terms of energy absorption.

In the current study, two temperature levels of 55 °C and 90 °C were adopted, since T_g_ is close to 120 °C. To obtain a homogeneous level of sample heating, a special heating chamber was used, as shown in Figure 2.

Figure 10 presents stress–strain graphs for two series of samples 5_1–5_3 and 6_1–6_3. For ambient temperature and pressure of 1 bar, the maximum stress level was obtained at the level of 320–340 MPa. At 55 °C, the maximum stress range was from 203 to 213 MPa, while at 90 °C it was from 143 to 160 MPa. Thus, for a temperature of 90 °C, the strength loss is about 50%. The results show that the TBC applied on the PMC substrate cannot be subjected to impact loads at elevated temperatures. Therefore, it is necessary to analyze the use of another material for stiffening the ceramic mat.

The graphs obtained in Figure 10a are similar in shape to the graphs in Figure 6b. Their common feature is the lack of a horizontal section after reaching the maximum stress. This indicates the appearance of significant damage, but not of a catastrophic nature. This can be seen in Figure 11 for all samples in the analyzed series 5. The outer edges of the TBC barrier were partly damaged.

Moreover, the graphs in Figure 10b and in Figure 6c are similar too. They are characterized by a linear, quite rapid stress drop after reaching the maximum value. This type of chart indicates the catastrophic effects of dynamic loading, and this is confirmed in Figure 12, where the TBC layer residues are visible for samples from series 6, heated to 90 °C.

## 5. Numerical Studies

Numerical tests are a valuable source of information, not only about the effort of the sample material subjected to high-speed compression, but also about the impact of various geometric parameters, heat flows, and friction coefficients [32,33]. Analyzing the literature on the PMC materials, the authors rarely decide to carry out numerical simulations. The reason for this is the complex internal structure of the composite and various damage models, such as delamination, buckling, and cracking of fibers and resin. In the global model it is difficult to include these types of phenomena, which is why the authors limit themselves to models without damage [14]. In [10] almost the same response of the sample material was obtained as in the SHPB test. The reason for the difference may be geometric imperfections: the sample was not perfectly cubic, and the surfaces in contact with the bars may not have been perfectly parallel. In [11], the authors paid attention to the friction parameter. In their simulations they used the value of 0.1 referring to experimental data; however, they also reported their initial research, where the friction coefficient varied in the range of 0.1 to 1, and the results changed by 6%. Friction reduction can be achieved by applying grease to the sample surface; its impact can also be taken into account analytically, as in [34]. In [19], a geometrically advanced RVE model with additional reinforcement in the z axis is shown; however, the authors point out technical problems with contact. Initially, the model did not have a matrix, but a small amount was introduced to make the simulation successful. It should be noted that the matrix model is geometrically complicated, because it must fill the space between the curved fibers; hence, it is also necessary to use elements of the tetragonal type. The matrix and fibers were fully modeled by the authors in [23]. The matrix was treated as elastic plastic material. Ductile damage and shear damage were used. Surface based cohesive behavior was also used to account for delamination on interfaces, [35,36,37,38,39,40,41,42,43]. However, there is a lack of material data, information on mesh density, and calculation time.

In current numerical studies, the goal was to determine the effort of the material for the range in which the sample in the laboratory test was not damaged. The load and geometrical dimensions were adapted, as they were for sample 2_1 (Table 1). The scheme for fixing the sample between the bars is shown in Figure 13.

The diameters of all bars were 20.5 mm, the Young’s modulus was 210 GPa, and the density was 7800 kg/m^3^. The projectile length was 0.4 m; the length of both input and output bars was 1.5 m.

The material for the TBC layer was treated as linearly elastic; the Young’s modulus was adopted on the basis of previous studies, and was 5.2 GPa; and the Poisson’s ratio was 0.26. C3D8R elements were used to build the TBC layer in the amount of 3456; the global element size was 0.5 mm.

The PMC composite used the “lamina” material model. The data that were used are shown in Table 2.

The substrate was made of 4 layers of laminate, 0.23 mm thick each. The model used elements of the SC8R “continuum shell” type (Figure 14) in the amount of 2304. To observe the effort level of substrate, the Tsai–Hill criterion was used.
(10)$$\frac{\sigma_{11}^{2}}{X_{1}^{2}}-\frac{\sigma_{11}\sigma_{22}}{X_{1}^{2}}+\frac{\sigma_{22}^{2}}{X_{2}^{2}}+\frac{\tau_{12}^{2}}{S^{2}}=1,$$
where:*σ*_11_—the normal stresses in the first direction;*σ*_22_—the normal stresses in the second direction;*τ*_12_—the shear stresses;*X*_1_—the tensile strength in the first direction;*X*_2_—the tensile strength in the second direction;*S*—the shear strength.

If the value of 1 is reached in Criterion (10) the material, of the element will be completely efforted and completely degraded.

Numerical studies were carried out in the Abaqus/Explicite program. General contact was used with a friction coefficient of 0.1. The first step was to compare numerical and laboratory results of the incident, reflected, and transmitted waves (Figure 15). In general, a good agreement between the measured results and finite element results was observed.

Figure 16 presents Huber–von Mises–Hencky (H–M–H) reduced stress distributions for the TBC layer. From the “input bar” side (Figure 16b), the values are irregular: the maximum value is 300.2 MPa and the minimum value is 267.7 MPa. This type of stress distribution indicates that the corners of the sample will be damaged initially, and the middle part, which is the least efforted, will be the last to be damaged. This type of phenomenon is caused by: the shape of the sample, the occurrence of friction, and the different state of deformation of the material. The material closer to the edge of the sample may deform more freely. This can be seen on the sample after static compression (Figure 5), where the TBC layer was squeezed out of the substrate. An irregular manner of sample damage, starting from its outer edges, was also observed in experimental tests for 1.5 bar and 2 bar pressure.

The H–M–H stress distribution in the TBC side is more regular. Despite this, the composite layer in contact with TBC (layer 1) shows effort dispersion from about 0.37 to 0.53 (Figure 17). In this case, the dominant effect is the shape of the sample and heterogeneous deformation. Effort in the two inner layers of the composite can be considered as homogeneous; however, the outer layer in contact with the “output bar” (layer 4) has irregular effort in the range of 0.25 to 0.43. In this case, the dominant phenomenon is the friction between the steel bar and the PMC.

## 6. Conclusions

The paper presents results for 18 samples: statically loaded (three samples), dynamically loaded at room temperature (nine samples), and dynamically loaded at elevated temperatures (six samples). The analyzed TBC layer can protect the PMC substrate against thermal shocks; however, our research shows that one should be careful in their engineering applications with thermal and mechanical loads. The following conclusions were drawn from the work we carried out:The technology presented in this work is not expensive and can be used to protect large composite objects.The dynamic load limit that the PMC/TBC structure may undergo has been determined. It corresponds to a pressure of 1 bar (strain rate ≈1500 1/s) and an impact speed *V*_0_ in the range of 7.73–8.26 m/s.Increasing the operating temperature of the PMC/TBC system to 90 °C results in a reduction of the dynamic strength of the protective coating by about 50%. Therefore, further research should be carried out for other materials stiffening the ceramic mat.The numerical model was made in the Abaqus program. Its version without damage description successfully predicted the elastic behavior of the PMC/TBC structure. The results obtained in the numerical simulation are consistent with the laboratory tests. The small difference in comparison to the experiments was due to imperfectly cubic samples’ geometry and lack of an ideal contact between bars and sample surfaces. A numerical model will be developed to include gradual degradation of the PMC/TBC structure under impact to the final failure.The FEM simulation allowed for a detailed determination of the effort of the substrate material and the protective layer. The stress in the protective layer was 300.2 MPa, while the effort in each layer of the composite was very low; i.e., did not exceed 53%.The reduced H–M–H stress distributions obtained in FEM simulation are consistent with images of damaged samples after laboratory tests. The middle part of the sample has the least effort, and the material begins to be damaged from the sample edges.

## Figures and Tables

**Figure 1 materials-13-00167-f001:**
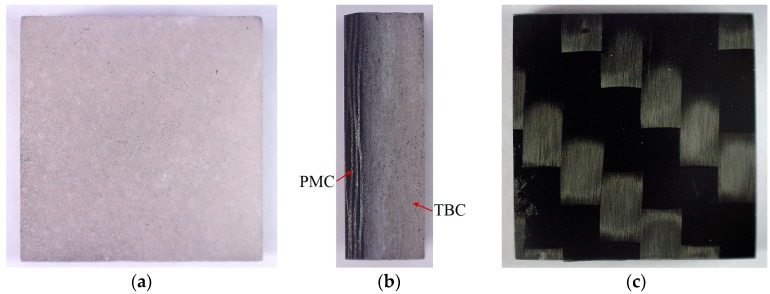
Polymer matrix composite (TBC)/polymer-matrix composite (PMC) sample: (**a**) the TBC ceramic mat protective layer, (**b**) cross section of the TBC/PMC system, and (**c**) the PMC layer.

**Figure 2 materials-13-00167-f002:**
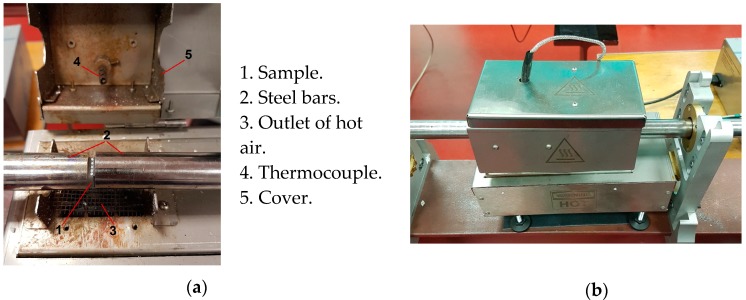
Tests using an electric furnace: (**a**) specimen sandwiched between the bars; (**b**) before the test.

**Figure 3 materials-13-00167-f003:**
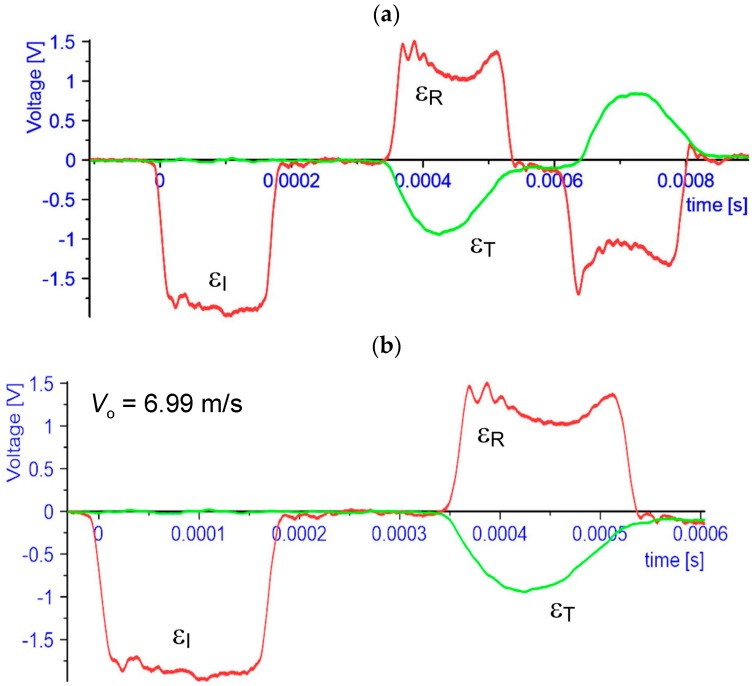
Results from SHPB (sample 6_3). (**a**) before cut (**b**) after cut.

**Figure 4 materials-13-00167-f004:**
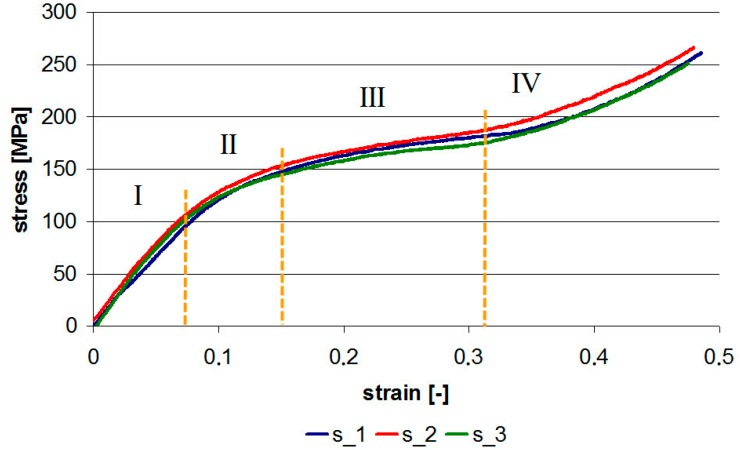
Stress–strain graph for static compression at constant strain rate of 4.16 × 10^−3^ 1/s.

**Figure 5 materials-13-00167-f005:**
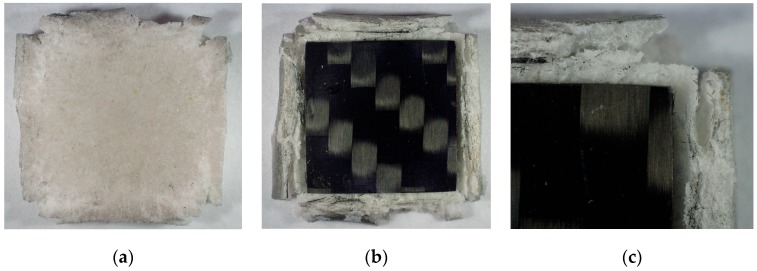
Sample 1_1 after static compression test: (**a**) the TBC ceramic mat, (**b**) the substrate PMC layer, and (**c**) details of damage stage after 2 mm displacement.

**Figure 6 materials-13-00167-f006:**
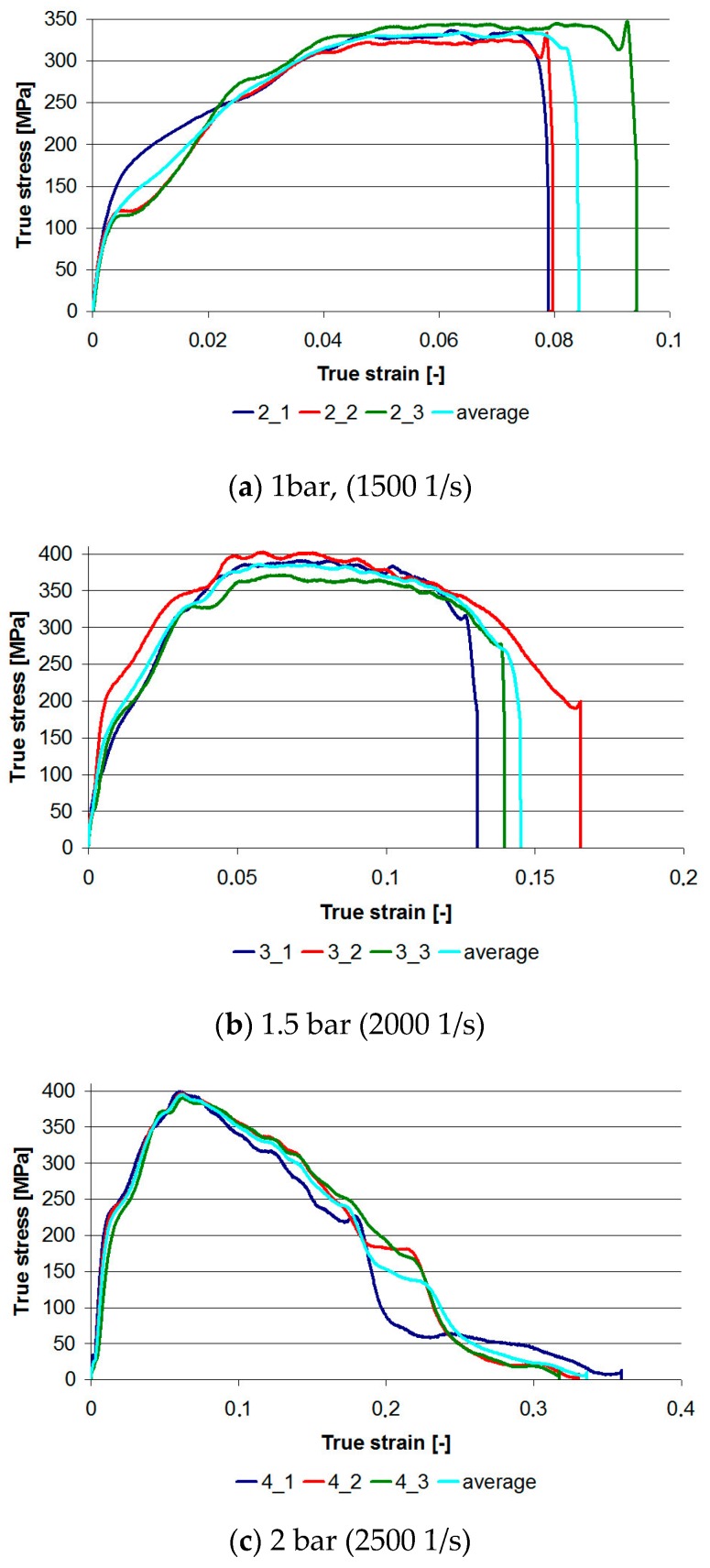
True stress–true strain graphs for three loads in dynamic compression tests at ambient temperature. (**a**) 21 °C—samples 2_1 to 2_3 (**b**) 21 °C—samples 3_1 to 3_3 (**c**) 21 °C—samples 4_1 to 4_3.

**Figure 7 materials-13-00167-f007:**
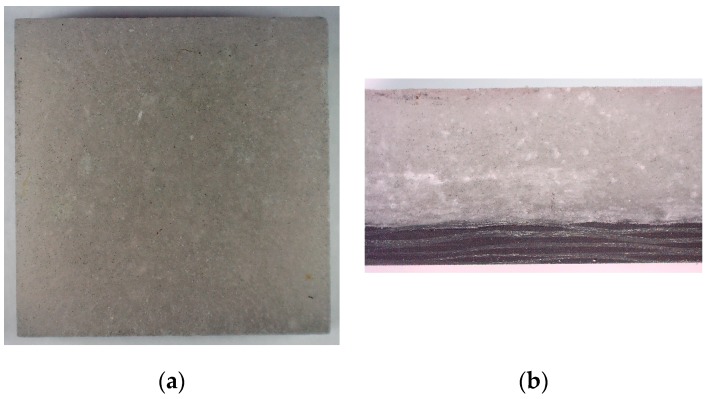
Internal structure of the TBC/PMC sample after impact with the pressure of 1 bar: (**a**) the TBC ceramic mat; (**b**) cross-section of the sample.

**Figure 8 materials-13-00167-f008:**
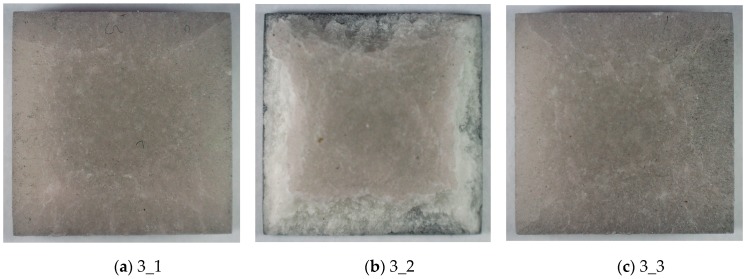
The TBC ceramic mat of samples after impact with the pressure of 1.5 bar. (**a**) sample 3_1 (**b**) sample 3_2 (**c**) sample 3_3.

**Figure 9 materials-13-00167-f009:**
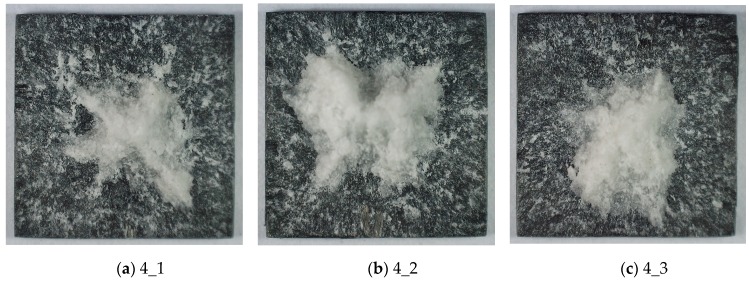
Failure of the TBC ceramic mat of samples after impact with the pressure of 2 bar. (**a**) sample 4_1 (**b**) sample 4_2 (**c**) sample 4_3.

**Figure 10 materials-13-00167-f010:**
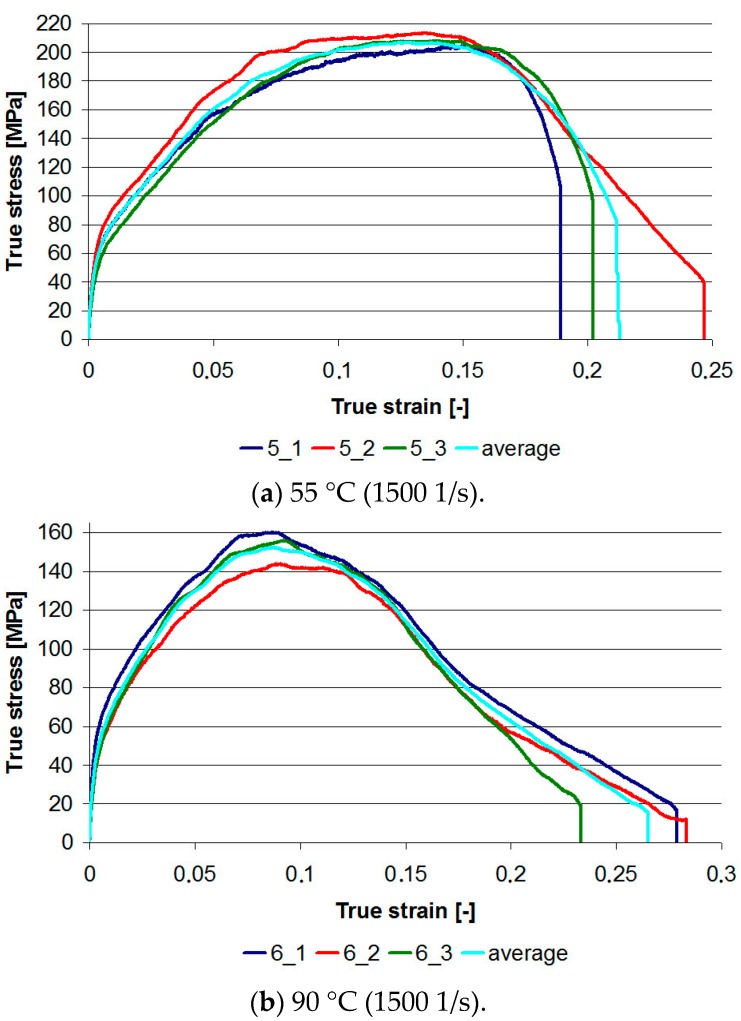
True stress–true strain graphs in dynamic compression tests for two temperatures: (**a**) 55 °C—samples 5_1 to 5_3; (**b**) 90 °C—samples 6_1 to 6_3.

**Figure 11 materials-13-00167-f011:**
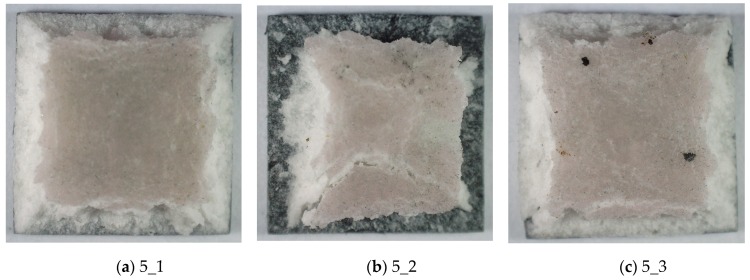
The TBC ceramic mats of samples after compressive impact tests at 55 °C. (**a**) sample 5_1 (**b**) sample 5_2 (**c**) sample 5_3.

**Figure 12 materials-13-00167-f012:**
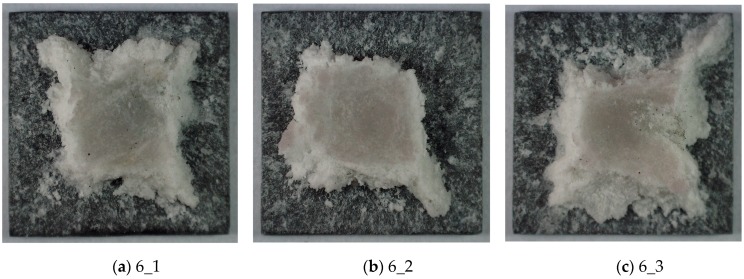
The TBC ceramic mats of samples after compressive impact the tests at 90 °C. (**a**) sample 6_1 (**b**) sample 6_2 (**c**) sample 6_3.

**Figure 13 materials-13-00167-f013:**
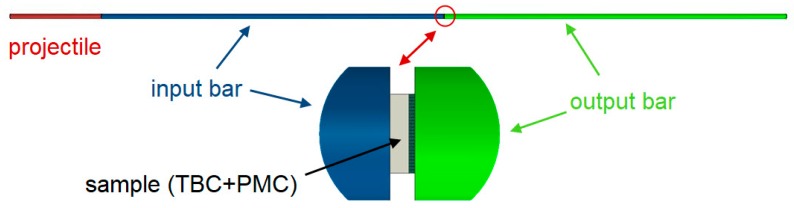
Scheme of sample mounting.

**Figure 14 materials-13-00167-f014:**
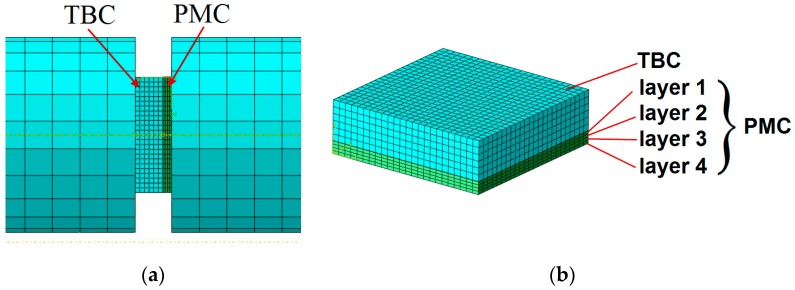
Mesh of finite elements. (**a**) sample between bars (**b**) layers of sample.

**Figure 15 materials-13-00167-f015:**
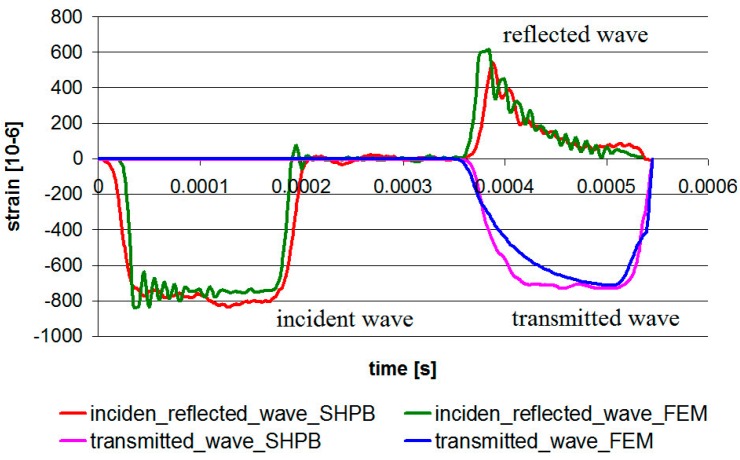
The comparison between numerical and laboratory results.

**Figure 16 materials-13-00167-f016:**
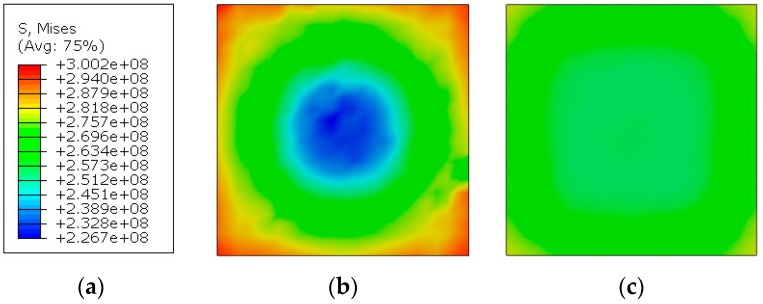
H–M–H stresses in TBC: (**a**) scale in [Pa]; (**b**) view from the “input bar” side; (**c**) view from the PMC side, above the layer 1.

**Figure 17 materials-13-00167-f017:**
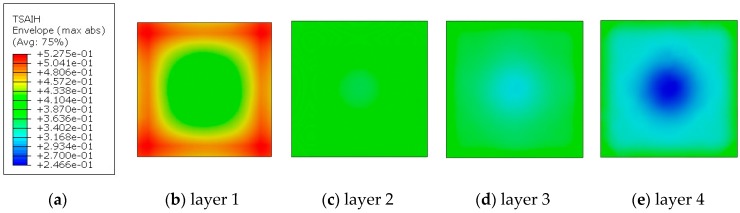
Tsai–Hill criterion for the PMC layers. (**a**) Tsai-Hill criterion (**b**) layer 1 (**c**) layer 2 (**d**) layer 3 (**e**) layer 4.

**Table 1 materials-13-00167-t001:** Characteristics of laboratory test.

No.	Dimensions [mm]	Pressure [bar]	Initial Impact Velocity *V*_0_ [m/s]	Temperature *T*_0_ [°C]
1_1	12.11 × 12.12 × 3.87	Quasi-static test		21
1_2	12.18 × 12.22 × 3.96	Quasi-static test		21
1_3	12.17 × 12.21 × 4.00	Quasi-static test		21
2_1	12.12 × 12.13 × 3.84	≈1	7.730	21
2_2	12.14 × 12.28 × 3.91	≈1	7.717	21
2_3	12.12 × 12.12 × 3.86	≈1	8.266	21
3_1	12.11 × 12.03 × 3.90	≈1.5	9.765	21
3_2	12.14 × 12.16 × 3.87	≈1.5	10.604	21
3_3	12.25 × 12.14 × 3.95	≈1.5	9.777	21
4_1	12.16 × 12.16 × 3.93	≈2	11.848	21
4_2	12.11 × 12.16 × 3.94	≈2	11.848	21
4_3	12.11 × 12.14 × 3.99	≈2	11.844	21
5_1	12.12 × 12.08 × 3.84	≈1	7.347	55
5_2	12.09 × 12.05 × 3.81	≈1	8.179	55
5_3	12.13 × 12.12 × 3.86	≈1	7.658	55
6_1	12.03 × 12.11 × 3.85	≈1	7.558	90
6_2	12.16 × 12.17 × 3.85	≈1	7.319	90
6_3	12.14 × 12.14 × 3.80	≈1	6.994	90

**Table 2 materials-13-00167-t002:** Elastic and strength properties for substrate.

E_1_ [GPa]		E_1_ [GPa]		ν_12_ [-]		G_12_ [GPa]	
55.5		55.5		0.04		3.00	
X_t_ [MPa]	X_c_ [MPa]		Y_t_ [MPa]		Y_c_ [MPa]		S [MPa]
828	580		828		580		105

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
