# Peer review of "Material Characterization of PMC/TBC Composite Under High Strain Rates and Elevated Temperatures"

_materials, 2020, doi:10.3390/ma13010167_

Round 1
Reviewer 1 Report
Material characterization of PMC/TBC composite under high strain rates and elevated temperatures by Golewski et al. analyzes the dynamic and thermo-dynamic mechanical behavior of PMC/TBC system under static and dynamic tests at room and elevated operating temperature. The manuscript provide useful and novel information. The experiments are well-designed and the results are well-presented. Overall, manuscript present a very coherent structure, in which the figures also contribute. It should thus be published in Materials.
Author Response
Reviewer 1
Re: We improved English according to Reviewer suggestion.

Reviewer 2 Report
Line 171: There are --> In the article, several times “there” is used to express results. It is requested to optimize the English writing.
Line 177: conclusion --> must be conclusions?
Improve quality of microscopic images such as Fig 7b
Line238-239 It is mentioned other materials. It is advised to define examples
Line 330 statically, dynamically….not so well formulated
Line 337 Conclusion is not well formulated. It is requested to expand and improve conclusions, also comparing the results - specially the modeling - with other research results published in other articles.
Line 340: It would be expected some suggestions on materials could be integrated
Author Response
Reviewer 2
Line 171: There are – In the article, several times „there” is used to Express results. It is requested to optimize the English writing.
Re: Some changes were made to limit the word “there”
Line 177: conclusion – must be conclusions?
Re: The change was made.
Improve quality of microscopic images such as Fig 7b
Re: For now, we don’t have better microscope. This images were made by 5MPx digital camera.
Line 238 – 239. It is mentioned other materials. It is advised to define examples
Re: In the future, we will probably use lithium water glass in the next tests. It is not good to present now our idea for preparation of the next paper.
Line 330 statically, dynamically … not so well formulated
Re: The change was made.
Line 337 Conclusion is not well formulated. It is requested to expand and improve conclusions, also comparing the results – specially modeling – with other research result published in other articles
Re: The change was made. Some conclusions were added.
Line 340: It would be expected some suggestions on materials could be integrated
Re: In the future, we will probably use lithium water glass in the next tests. It is not good to present now our idea for preparation of the next paper.

Reviewer 3 Report
The research Golewski et al. «Material characterization of PMC/TBC composite under high strain rates and elevated temperatures» is devoted to the analysis of PMC/TBC systems, statically, dynamically loaded, at room temperature and at elevated operating temperatures.
The methodology realized in the manuscript is well described. The results are represented in detail and discussed. Taken as a whole, the research is interesting. It’s worth publishing in “Materials” journal.
There are some comments of mine given below:
- The information provided in the abstract is too general. It should be clarified in the context of which particular polymer composites (type of matrix and filler) the content of the work is described.
- Section "3. Materials and methodology" should specify the characteristics of the ceramic mat and hardener used.
- In Fig. 2, Fig. 5 and Fig. 11 the notation should be corrected.
- It is not clear what the corresponding notation s_1, s_2 and s_3 in Figure 4.
Author Response
Reviewer 3
The information provided in the abstract is too general. It should be clarified in the context of which particular polymer composites (type of matrix and filler) the content of the work is described.
Re: Information about substrate material and thermal barrier were added to the Abstract.
Section “3. Materials and methodology” should specify the characteristics of the ceramic mat and hardener used
Re: Both materials are commercial and made by Vitcas company. This company do not provide any information (trade secrets), therefore in works [6] we made some tests to determine strength and Young modulus of ceramic mat saturated by hardener.
In Fig. 2. Fig. 5. and Fig. 11 the notation should be corrected.
Re: Changes were introduced and notation was corrected.
It is not clear what the corresponding notation s_1, s_2 and s_3 in Figure 4.
Re: Three samples were subjected for static compression tests: 1_1, 1_2, 1_3. This comment was add to the text.
